# Evaluation of Growth Responses of Lettuce and Energy Efficiency of the Substrate and Smart Hydroponics Cropping System

**DOI:** 10.3390/s23041875

**Published:** 2023-02-07

**Authors:** Monica Dutta, Deepali Gupta, Sangeeta Sahu, Suresh Limkar, Pawan Singh, Ashutosh Mishra, Manoj Kumar, Rahim Mutlu

**Affiliations:** 1Chitkara University Institute of Engineering and Technology, Chitkara University, Rajpura 140401, India; 2Department of Chemistry, Bhilai Institute of Technology, Raipur 493661, India; 3Department of Artificial Intelligence & Data Science, AISSMS Institute of Information Technology, Pune 411001, India; 4Department of Computer Science, Central University of Rajasthan, Ajmer 305817, India; 5School of Integrated Technology, Yonsei University, Seoul 03722, Republic of Korea; 6Faculty of Engineering and Information Sciences, University of Wollongong in Dubai, Dubai P.O. Box 20183, United Arab Emirates; 7Intelligent Robotics & Autonomous Systems Co. (iR@SC), RA Engineering, Shellharbour, NSW 2529, Australia

**Keywords:** substrate cultivation, hydroponics, energy efficient, nutrient film technique, sustainable growth, agricultural productivity, soft sensing, IoT

## Abstract

Smart sensing devices enabled hydroponics, a concept of vertical farming that involves soilless technology that increases green area. Although the cultivation medium is water, hydroponic cultivation uses 13 ± 10 times less water and gives 10 ± 5 times better quality products compared with those obtained through the substrate cultivation medium. The use of smart sensing devices helps in continuous real-time monitoring of the nutrient requirements and the environmental conditions required by the crop selected for cultivation. This, in turn, helps in enhanced year-round agricultural production. In this study, lettuce, a leafy crop, is cultivated with the Nutrient Film Technique (NFT) setup of hydroponics, and the growth results are compared with cultivation in a substrate medium. The leaf growth was analyzed in terms of cultivation cycle, leaf length, leaf perimeter, and leaf count in both cultivation methods, where hydroponics outperformed substrate cultivation. The results of the ‘AquaCrop simulator also showed similar results, not only qualitatively and quantitatively, but also in terms of sustainable growth and year-round production. The energy consumption of both the cultivation methods is compared, and it is found that hydroponics consumes 70 ± 11 times more energy compared to substrate cultivation. Finally, it is concluded that smart sensing devices form the backbone of precision agriculture, thereby multiplying crop yield by real-time monitoring of the agronomical variables.

## 1. Introduction

Farming has been an integral part of many nations around the world for ages. It is a more productive version of vegetation that sustains life on Earth in many ways. On the other hand, urbanization has grown exponentially over the last few decades. Along with the opportunities and innovations that urbanization brings, the need to develop green areas in urbanized localities is rising rapidly. This is due to the urgent requirement to maintain a cleaner and healthier living environment for the inhabitants of the Earth. The widespread use of the Internet, which connects almost all components of the physical world to improve the quality of people’s lives, is evolving with the advent of the Internet of Things (IoT). Vertical Farming (VF) has proved to be beneficial in maintaining and expanding environmental sustainability, along with enhanced food security in rapidly growing urban areas. Limited available land for cultivation is emerging to be the biggest hindrance to farming in urban societies which is again solved by the adoption of IoT technologies in the concept of VF [1].

The concept of growing crops in an indoor environment was introduced to ensure year-round yield. The cultivation is carried out in vertical stacks to reduce cultivation space [2]. Providing optimal conditions required for the crops to flourish all year is also termed Controlled Environment Agriculture (CEA) [3]. Apart from facilitating the maximum possible yield in a small area, CEA mainly aims at achieving better crop quality in a sustainable manner using minimal resources, such as energy, area, manpower, investment, and water. Vertical farming has many advantages over conventional agriculture because it utilizes minimum resources and expenses while giving the best returns throughout the year, irrespective of seasonal dependencies; therefore, it is more sustainable. In traditional farming systems, crop yield is vulnerable to natural calamities, global warming, and climatic conditions. All these constraints cause lower yields and a lack of natural resources to fulfill the requirements of the ever-increasing population. Vertical farming has proven itself to be a highly effective way to solve these issues, but there are a few limitations to it too.

The three methods of artificial farming are aeroponics, hydroponics, and aquaponics. In the aeroponics method of vertical farming, the plants are suspended in air, and the lower parts of the stems and roots are sprayed with nutrient spray. In the hydroponics method of vertical farming, the cultivation medium is nutrient-infused water. The plants are supported with a supporting medium, such as cocopeat, and the roots are supplied with water dissolved with an adequate proportion of nutrients at specific intervals. In the aquaponics method of artificial cultivation, an ecosystem of aquaculture is maintained. Fish are cultured in a tank, and the water from the fish tank is supplied to the hydroponic setup. Plant compost is made from fish, and the water, thus cleaned, is recycled for the fish. This way, fish and plants thrive simultaneously in this ecosystem. All these cultivation methods practice soilless and much faster farming. Vertical farming along with precision farming is made practically feasible with the use of the IoT [4]. Amongst the varied and widespread applications of the IoT, their use in agriculture still has a lot to explore in terms of practical applications. The use of the IoT in traditional agriculture processes has been tried and tested but their use in vertical farming methods is yet to be explored to the fullest [5]. For faster and healthier growth of plants, aeroponics helps in attaining a pest-free and disease-free growing environment [6]. The parameters that generally need to be controlled are the pH level, moisture of the growing medium, and light and temperature of the environment. All the parameters except sunlight can be controlled manually. This natural resource is thus replaced by grow lights to carry out indoor farming in urban areas. Types of available grow lights are: light emitting diodes (LED) grow lights, fluorescent grow lights, and high-pressure sodium (HPS) or high-intensity discharge (HID) grow lights [7]. All the threshold values are set in the sensors, and if any parameter crosses the range or drops below a limit, alternative arrangements will be activated to compensate [8]. The agricultural setup is generally located in residential and commercial areas where the user may be able to attend to other commitments, leaving the system to attend to the crops [9]. A pictorial depiction of the evolution of vertical farming is represented in Figure 1.

The most basic human necessity is food, and the industry that has been responsible to provide food is agriculture. It is one of the most primitive practices that is associated with the traditions and practices of the related geographic region.

Traditional substrate farming can be defined as a method of growing crops in soil using traditional methods of sowing, fertilization, irrigation, harvesting, etc., that are mainly dependent on available natural resources, regional climatic conditions, outdated tools, and cattle. On the other hand, modernization has caused a complete turnaround in agriculture in which technology, improved tools, machinery, artificial means of monitoring and controlling the growing environment, etc., are used [10]. This has also caused a shift from the farming community to agribusiness, even for amateur farmers.

In vertical farming, dependency on the bare essential requirement—soil medium—is eliminated. Crops can be planted in vertical stacks which have water or air as the growing medium. With the increase in population and urbanization, farming land has decreased considerably, and the demand for food is increasing exponentially. Thus, vertical farming acts as a boon to urbanization and maintains the food supply as per demand sustainably [11,12,13].

The differences between traditional and vertical farming methods are shown in Table 1.

The following research questions were formulated to guide our research in this study:

RQ. 1: What is the most effective artificial method for the cultivation of romaine lettuce?

RQ. 2: Which hydroponic setup is the most convenient one for growing leafy vegetables?

RQ. 3: What is the comparative growth analysis for the growth of romaine lettuce in traditional and hydroponic methods?

RQ. 4: How relatable are the results of actual implementations concerning simulator results?

RQ. 5: Which method is more beneficial, keeping in mind the sustainability, cost-effectiveness, and productivity of the crop under consideration?

RQ 6: Which method is more effective in terms of energy consumption?

Though vertical farming methods are more effective than traditional methods of farming, there is a need to identify the most effective artificial method of growing a particular crop. Each of the vertical farming methods has various setups per the requirements of the crop under consideration.

This research makes key contributions to the field of knowledge by comparing the growth characteristics of romaine lettuce using the substrate and vertical farming methods by: (1) cultivating romaine lettuce in the soil during its growing season and monitoring its growth by providing necessary nutrients, micronutrients, pH level, etc.; (2) cultivating romaine lettuce in an NFT-based hydroponic setup and monitoring its growth while providing it with all necessary climatic conditions and nutrient requirements; (3) comparing the qualitative and quantitative growth trend of romaine lettuce in soil vs. in NFT hydroponics; (4) finding the growth trend of romaine lettuce in the ‘AquaCrop’ simulator with conditions set for soil as well as the NFT; (5) qualitatively and quantitatively comparing the romaine lettuce growth trend from the simulator; (6) comparing the results obtained from actual cultivation methods and the simulator; and (7) analyzing the energy effectiveness of both methods of cultivation in terms of electricity consumed.

This research work is organized as follows: Section 1 covers the introduction to vertical farming and its evolution. Section 2 covers an extensive background study and the present state of the art in the hydroponic perspective of vertical farming. The materials and methods used in substrate cultivation and smart hydroponic cultivation methods are mentioned, discussed, and depicted in Section 3, along with the input parameters, their respective values and ranges, and a flowchart representing the methodology of the research work. Section 4 contains the results and discussion where a comparative analysis of the output parameters of romaine lettuce is mentioned and discussed. The outcome of the study is discussed in terms of leaf length, perimeter, and count, and the actual results are compared with those obtained from the simulator. The energy effectiveness of both methods is compared and discussed. Finally, the conclusion of the research and its future scope are mentioned in Section 5.

## 2. Related Studies

An automatic IoT-embedded aeroponic system was designed which comprises a mobile app, a sensor containing the IoT, and a platform. These three components provide GUI for remotely managing the aeroponic setup, a middleware for the mobile app, and control for individual pumps and for gathering data. Raspberry Pi Zero is used here, and the system is modelled in a way to alter the climatic factors as per the requirements [16,17,18,19,20,21,22,23]. The IoT keeps track of humidity, temperature, nutrient solution, PH, and electrical conductivity in the hydroponic farming ecosystem. The system in [7,8,22] has the IoT keep track of parameters in a hydroponic farm which also aims to help amateur farmers by making the system automatic through an Android app and a built-in alarm that alerts the user of any abnormal condition. Similar alternative arrangements for hydroponics can also be seen in [10]. The authors proposed a prediction model that monitors real-time data of sensor nodes via a machine learning algorithm and the fall curve method [24,25]. Plant health can also be monitored remotely through the IoT [26,27,28,29]. Automation of the monitoring of the greenhouse environment and the PH level and electrical conductivity is focused upon in this work. Data are transferred to the Internet by the IoT, and real-time status is monitored and maintained easily by an application [15]. The paper focuses on connecting a smart hydroponic model to the Cloud server with the help of the IoT. The mobile or web application deployed displays the data [30,31,32,33,34,35]. All the parameters that facilitate the healthy growth of plants indoors are monitored and controlled in this system. Plants can grow much faster and healthier if all the parameters are within controlled limits [17]. An LED-equipped IoT system is developed in this smart hydroponic system. Software applications equipped with the IoT are used to transmit and project real-time data [18]. Tomato plants are monitored and controlled using the IoT and a sensor-enabled hydroponic setup. The sensors are interfaced with Arduino and Raspberry Pi3 which act as edge developments of intelligence at the edge, using a deep neural network model to provide correct and real-time limiting actions to hydroponic systems [19]. An IoT-based hydroponic system is built using sensors and the hardware needed. This integration is performed using a NodeMCU ESP8266 Wi-Fi Module [20]. An iHydroIoT—an IoT monitoring system for hydroponics—is proposed in this work. A prototype for the acquisition of data and a mobile app for iOS is developed. Plotly—a data analytics and visualization library—stores all the accumulated data. Humidity, light, CO_2_, temperature, electromagnetic conductivity, water level, and PH are monitored [21]. Smart irrigation using the IoT for hydroponic systems was proposed and a prototype was developed monitoring real-time data, such as temperature, water flow, and humidity. All this was controlled and logged with the aid of the ThinkSpeak IoT platform [22]. PlantTalk—an intelligent IoT-based hydroponic plant factory is proposed. The experimental results using PlantTalk intelligence reduced the CO_2_ concentration at a rate 53% faster than the traditional system. AgriTalk—a plant factory—has used PlantTalk. A plant-care box and Devil’s Ivy have also used PlantTalk as an example to show how the hydroponic factory works. Humidity, pH, CO_2_, temperature, O_2_,and H_2_O sensors are used in this system along with many soft control buttons and a timer [23]. A hydroponic agriculture precision and monitoring system based on fuzzy logic and the IoT concept was proposed. Plants are monitored for their nutrient level and other necessities using the IoT, and the supply of water and nutrients is controlled using fuzzy logic. Lettuce and bok choy were experimented with using a smart hydroponic setup based on fuzzy logic in [19,21] and were found to grow better in terms of leaf size in [16,36,37,38,39]. Raspberry Pi and Arduino are used to monitor the various parameters that are known to affect crop yield, such as CO_2_, moisture in the growing medium, humidity, temperature, and light intensity. Visualizing on the ThingSpeak platform and the real-time data collection from Smart was found to give an enhanced yield and 100% success rate, keeping the parameters in the correct range [40]. Compared to conventional farming methods, lettuce leaves were found to grow over 40% larger, with more robust roots and density. Approximately 400% higher performance was recorded [41]. The plants showed temporal variations and light intensity variations, along with in-range humidity, pH, temperature, electrical conductivity, and CO_2_. To ensure improved supervision in the case of hydroponics, the system also monitored the water level of the container [42]. A unique system for hydroponic cultivation was proposed in Europe which has two greenhouses and an LED-lit phytotron [43]. The authors in [4] have come up with a GUI for monitoring the aeroponics setup, sensors to control the individual pumps, and middleware for the mobile application. Raspberry Pi Zero is used in the automatic system which can alter climatic factors when needed [28]. A Cloud server along with Open Garden and a Wi-Fi module was used in [9,27,44], and one with edge computing was detailed in [23]. A comparison of blue and red LEDs checks for better efficiency in a smart hydroponic system [11,26,45].

A hybrid Wi-Fi–Zigbee technology, along with WSN to evaluate the system performance error was developed in [46]. Deep neural networks and RNN-LSTM for prediction were used by the authors in [24]. Various ML algorithms, such as KNN, lasso regression algorithm, and random forest algorithms, were used by the authors in [25,26], and the random forest algorithm was found to be 90.62% accurate with the sensor fusion concept [6]. Authors in [29] showed that hydroponically grown crops in a controlled environment are healthier and take lesser time to yield than those grown traditionally.

A deficit of micronutrients, such as calcium, magnesium, sulfur, and potassium, in a hydroponic environment, using image processing was carried out concerning chili, combining the factors of color, shape, and texture of the leaves was proposed in [31]. Figure 2 shows a graphical representation of the present state of the art of hydroponic farming.

## 3. Materials and Methods

### 3.1. Cultivation of Romaine Lettuce (Lactuca sativa *L*. var. longifolia)

Romaine lettuce is the subject crop selected for this study because it is a leafy crop that thrives well in a soil as well as a water medium. The study mainly involves the cultivation of romaine lettuce in soil, i.e., the substrate cultivation method, and in a hydroponic setup, specifically, the nutrient film technique.

The soil cultivation must be carried out at a specific time of the year when the weather conditions are most suitable for the crop. In contrast, the hydroponic method of cultivation is a kind of vertical farming where the environmental conditions as well as the nutritional requirements of the crops are provided externally in artificial ways, thus ensuring year-round production.

The NFT on the other hand is a specific setup of hydroponic cultivation whereby a grow tube with numerous holes is supplied with nutrient solution continuously. The plants are held in perforated cups and held by some supporting medium. All the conditions, such as pH, temperature, TDS, etc., required for the growth of the specific crop are provided externally.

#### 3.1.1. Cultivation System: Substrate Cultivation

The experiment was carried out in a nursery bed from August to October. Around 260 L of water was consumed to irrigate per kg of romaine lettuce. The nutrient content of the soil maintained per acre of cultivated land was: 23.6 kg of nitrogen (N), 158.7 to 190.5 kg of phosphorous (P_2_O_5_), and 92.25 kg of potassium (K_2_O). Apart from these, calcium (Ca), magnesium (Mg), boron (B), and other micronutrients were provided to the cultivated land. The pH content of the soil must be within the range of 5.6 to 6.1. Table 2 shows the parametric values and ranges in the substrate cultivation method.

Figure 3 shows a picture of the setup of a traditional method of substrate agriculture.

#### 3.1.2. Cultivation System: Hydroponic Method Using Nutrient Film Technique

The experiment was carried out in a polyhouse. It was carried out in September, but it may be carried out at any time of the year. Two tanks, Tank A and Tank B, were needed for the setup. Tank A was used for 2 to 3 days, and then tank B was used. The NFT setup of hydroponic farming was used.

Tank A contained iron (Fe), calcium (Ca), magnesium (Mg), and 10 L of stock solution. Tank B contained MICRO–NPK stiffness phosphorous for root growth. Nitric acid was used to reduce pH and keep it in control. Hogland solution was used, and pH was maintained between 5.5 to 6.5, whereas TDS was maintained between 102 and 301. Figure 4a is a picture of a real NFT setup with romaine lettuce at the initial phase of growth, whereas Figure 4b gives us a glimpse of an NFT setup with romaine lettuce thriving in it.

As this is a man-made setup and production continues throughout the year, temperature conditions must be maintained at a permissible level of 25 to 30 degrees Celsius, and the total water consumption is 20 L/Kg of the produce.

Three measuring cylinders with EC 0.6–0.7, 6.2 pH, EC 0.5, 7.4 pH, and EC 0.8 and avg pH 6.1 were used, where the EC limit was the limit of the electric conductivity above which the nutrient solution was completely replaced. HCL and nitric acid were among the other essential requirements. A water meter was used to measure the water consumption in this process. Table 3 is another tabular representation to project the parameters and their ranges in an NFT hydroponic setup.

Figure 5 shows the measure of TDS in the water solution. Therefore, research into determining the response to RQ. 2 leads to the answer that the NFT is the most convenient hydroponic setup for growing leafy vegetables.

#### 3.1.3. Methodology

The workflow of the research work is depicted in Figure 6, which begins with the selection of the subject crop. The crop selected here was romaine lettuce or *Lactuca sativa* L. var. *longifolia*. This crop was grown in a substrate medium as well as in an NFT hydroponic setup simultaneously. The flowchart in Figure 6 shows that the time taken for the lettuce to fully grow in the soil was 60 to 90 days and that in the hydroponic cultivation method it was only 30 to 40 days. After harvesting the crop in both the mediums, qualitative analysis, such as leaf length, perimeter, and count, as well as quantitative analysis, such as production per unit area of the cultivation medium, amount of water required, etc., was carried out. Finally, the obtained results were verified with the results obtained from the ‘AquaCrop’ simulator, and the outputs of both methods of experimenting were verified.

#### 3.1.4. Block Diagram

Figure 7 shows the process of remote sensing and controlling smart farms with the IoT. Substrate cultivation as well as vertical farming methods can be integrated with the IoT to make them smart. This, in turn, helps in remote communication with the user, database, and Cloud server. Sensors and actuators help in controlling the farms remotely in response to the analysis performed in the Cloud. Smart farmers can use their desktops or smartphones to access the details of their smart farms remotely. Sensor data stored and processed in the Cloud also ensure the security, integrity, and reliability of the raw data as well as the processed result.

In this research, the authors have proposed a smart hydroponic setup that operates on the concept of renewable energy obtained from natural resources. Energy-efficient consumption is exhibited by using a solar panel for the entire unit to run efficiently. The proposed smart hydroponic system is explained in Figure 8. There are three basic blocks in the entire setup: a smart hydroponic setup, an intermediate microcontroller connected to the Cloud and energy source, and a hardware module with all the associated actuators. The smart NFT hydroponic setup is equipped with a tank, motors, timer, temperature sensor, humidity sensor, pH sensor, and O_2_ and CO_2_ sensors for continuous monitoring of the required parametric ranges. The input values are read by the microcontroller and sent to the Cloud for analysis, and the actuators are used to activate and deactivate the hardware devices in the actuator module.

The actuator module consists of a fan, cooler, water pump, nutrient pump, mist spray, LED, and buzzer.

The dissolved oxygen content of water is essential for plant growth. It tends to decrease over time; therefore, the old water in the tank or the reservoir must be flushed out and refilled monthly. Although most of the roots must be submerged in water, roots must be partially exposed to air in some intervals.

The system can be broadly classified into two sections, namely, hardware components that process and control the system and software components that update sensor values and actuator status. The proposed smart hydroponic setup has several hardware and software components as shown in the block diagram in Figure 9.

The hardware components consist of sensors and actuators. Sensors are used to sense the various parametric conditions to keep a check on the system. Various sensors, such as pH sensors, water sensors, air temperature, and humidity sensors, are used in this setup. Each of the sensors is set with specific parametric ranges, and the values obtained are monitored continuously by the microcontroller. Actuators, on the other hand, are the hardware components that facilitate making the alternative arrangements operate according to the values sensed by the sensors and analyzed by the microcontroller. If the value of any parameter exceeds or drops from the defined range, alternative arrangements are activated and deactivated according to the need of the situation. Thus, allowing the vertical urban farmers of to attend to their other commitments while leaving the machines to take care of the system. Figure 10 shows the experimental setup consisting of some of the actuators and sensors.

Various sensor modules and other hardware components used in this model can be described as:**Temperature module:** The temperature is set in the range of 25 to 30 degrees Celsius which is the most suitable range for lettuce cultivation. The microcontroller fetches and analyses the temperature values from the temperature sensor: (i) If the temperature exceeds 30 degrees Celsius, there are provisions for two exhaust fans and coolers in wet walls in the polyhouse. The temperature is maintained by the operation of these devices and continuous monitoring. As soon as the temperature range is reached, the exhaust fans and coolers are turned off. (ii) No action was taken in this experiment when the temperature was low. This is because this experiment was carried out in September when the temperature does not fall below the tolerable limits for lettuce. Romaine lettuce can tolerate a minimum temperature of up to 18 degrees Celsius. In conditions where the temperature is projected to fall below this level, alternative arrangements for heaters can be set up.**pH module:** The pH value of the pH sensor is set to be between 5.4 to 6.6. The value is fetched by the microcontroller and the following conditions are checked: (i) If the pH value falls below 5.4, pH high solution is added to the water medium until the pH comes within the permissible range. (ii) If the pH value rises above 6.6, pH low solution or nitric acid is added to maintain the balance.**Humidity module:** The microcontroller fetches the humidity values from the humidity sensor: (i) If the humidity is found to be low, mist sprinklers are activated to bring the values within the needed range. The mist sprinklers are turned off as soon as the humidity reaches the set range. (ii) If the humidity is higher than the threshold value, fans are operated for the circulation of dry air.**Water level sensing module:** This sensor module helps determine the water level in the grow tubes. If the amount of water in the grow tubes is nil or below a threshold value for more than a set period, the water pumps are activated and nutrient-rich water starts flowing into the grow tubes. It indicates when a certain quantity of water is to be supplied by the pumps to the system.**CO_2_ and O_2_ level modules:** These sensor modules check the CO_2_ and O_2_ levels in the hydroponic setup. This helps in knowing how efficient the crop is in terms of refreshing the surrounding air. Whenever the level of CO_2_ reaches or crosses the range of 1000 ppm or the level of O_2_ falls below 18%, air purification is processed.**LED and Buzzer:** Light-emitting diodes and a buzzer are the hardware units connected to the smart hydroponic system under study. They notify the user of any change in the existing condition that must be investigated, either automatically or manually.**Timer:** A timer is set in connection with the water and nutrient pumps for their routine functioning.

## 4. Results and Discussions

### 4.1. Actual Cultivation Results: Substrate Cultivation vs. Hydroponics

Comparative analysis between romaine lettuce grown in substrate media and that grown hydroponically is carried out in this section. The growth in both types of mediums is compared under various parameters, and the results are compared in Table 4.

The complete harvesting time taken for romaine lettuce to thrive fully in soil medium is 60 to 90 days, and the productivity obtained was found to be 3.9 kg per square meter of cultivated land. The maximum recorded length of leaves was 5 to 7 cm and the maximum recorded perimeter of the leaves was 10–20 cm. Soil medium is not devoid of pests, rodents, and other organisms damaging the crops. Even after treatments, the estimated number of infected plants was 20–40 plants out of 100 plants. One of those plants is shown in Figure 11a.

Figure 11b is a picture of a healthy romaine lettuce plant grown hydroponically. In the case of hydroponically grown romaine lettuce, the setup was performed according to the NFT, the complete harvesting time to thrive fully in water and nutrient solution medium was 30 to 40 days, and the productivity obtained was 41 Kg per square meter of cultivated land. The maximum recorded length of the leaves was 8 to 10 cm, and the maximum recorded perimeter of the leaves was 20 to 30 cm. Even though water is a medium that cannot sustain pests and rodents, there remains a possibility of fungus infections in the crop. If treated timely with fungicides, disease infestation can be as low as 5 to 15 out of 100 plants, which is negligible compared to that in soil. In the experiment conducted in hydroponics, the number of infected plants recorded was around 6 out of 50, whereas in the case of substrate cultivation, the number of infected plants was around 17 out of 50. The weather requirements as well as the nutrient requirements are completely nature-dependent in the case of substrate cultivation methods.

Another dimension of this research is the analysis of the energy efficiency of the substrate and hydroponic methods of cultivation. In this analysis, the substrate cultivation method was found to be more energy efficient than the hydroponic method.

A comparative yield analysis corresponding to various parameters is shown in Table 4.

### 4.2. ‘AquaCrop’ Simulator Results in Substrate Cultivation

A simulator named ‘AquaCrop’ was used to analyze and verify the results obtained from the actual planting results in soil. The simulator was fed various parametric values, such as minimum and maximum temperature, the period taken by the crop to grow fully, the specific months when the crop thrives in soil, the type of soil, amount of water needed for irrigation, rainfall measures, etc. The results obtained in terms of canopy cover (CC) of romaine lettuce are depicted in Figure 12.

Figure 12 is a clear depiction of the CC of the yield. Given all the suitable environmental conditions, there are chances of less yield in the case of the substrate method of cultivation due to many factors, such as weeds, microbes, pests, and rodents, in the soil. Thus, the yield is hampered to some extent. The light green portion of the CC shows the fall in yield in the soil medium. The period is set from August to October, whereas the biomass obtained was 2.326 tons/ha and the dry yield obtained was 1.977 tons/ha.

### 4.3. ‘AquaCrop’ Simulator Results in Hydroponic Cultivation

Figure 13, on the other hand, is the simulation output obtained when the hydroponic conditions are set in the simulator. The temperature, the growth cycle, the type of medium, the amount of water needed, etc., were fed to the simulator. The output results in terms of CC are visibly higher than that in the case of substrate cultivation, thus giving a better yield in less than half the period. A substantial increase in biomass, as well as dry yield of the produce, is also noticed in the case of hydroponics. Therefore, RQ. 3 can be answered as the hydroponic method of lettuce cultivation outperforms the substrate method of cultivation. Table 4 is a tabular representation of the quantitative comparison of lettuce production with the substrate method against that produced with the hydroponic method.

### 4.4. Comparison of ‘AquaCrop’ Simulator Results: Substrate Cultivation vs. Hydroponics

Table 5 is a tabular representation of the comparative analysis obtained from the simulation results.

To sum up the results of the simulator, it can be said that though the cultivation time is twice as long in the case of substrate cultivation, the quality of the yield in terms of CC, as well as the quantity of yield in terms of biomass and dry yield, is much less compared to that obtained hydroponically. Thus, in response to RQ 1, we can state that the most effective artificial method for the cultivation of romaine lettuce is NFT hydroponics.

### 4.5. Comparison and Verification of Actual Results with the Simulator Results

Figure 14 is a graphical representation of the maximum length of romaine lettuce leaves obtained in soil and with the hydroponics method. It is also a clear depiction that substrate cultivation takes more than double the number of days to grow fully but still produces smaller leaves compared to hydroponically grown lettuce, which takes less time to grow and produces bigger leaves. The x-axis is labeled as the length of the leaves in cm, and the y-axis shows the cultivation time of hydroponics and the substrate method of cultivation.

A periodic plot of the leaf length of romaine lettuce grown in soil as well as in hydroponics is depicted in Figure 15. The x-axis is labeled as the number of days from sowing the plant in a periodic interval of 10 days, and the y-axis is labeled as the leaf length in cm. This depiction clarifies the rate of growth of a leaf in both mediums at an interval of 10 days from the day of sowing. Another inference that can be derived from this plot is that in addition to taking less time to grow fully, the pace of growth of a hydroponic lettuce leaf is double that of the leaf grown in soil.

To compare the leaf perimeter of romaine lettuce in soil and hydroponic medium periodically, a graph has been plotted in Figure 16, which shows that the leaf perimeter of the hydroponic crop is more than seven times that of the yield cultivated in soil in half the time, thus providing another aspect of leaf health that is seen to be improved in the hydroponic setup.

On the other hand, Figure 17 is a graph plotted to analyze the leaf production in terms of leaf count. It can be derived that the leaves of hydroponically grown romaine lettuce are not only healthier but also more abundant compared to the substrate method of cultivation. The leaf count reached almost double in the case of hydroponics compared to the soil cultivation method in half the time.

Based on studies as well as on a real experiment on growing romaine lettuce in the substrate as well as in the NFT setup of the hydroponic cultivation method, the hydroponic method of cultivation outperforms the substrate method. The amount of water consumed by the hydroponic setup was 20 L/kg of the product, whereas that consumed by the cultivation method in soil amounted to 260 L/kg of the produce. Therefore, the amount of water needed for the hydroponic cultivation of lettuce is almost 13 times less than that needed for substrate cultivation. In addition, the dependency on natural resources is eliminated. The pH, temperature, N, P, K, Fe, Ca, Mg, and other nutrient content of the cultivation medium is controlled according to the required conditions. All these factors lead to a two times better yield in the hydroponic method, both qualitatively and quantitatively.

The advantages of the proposed hydroponic system can be shown as:Better growth of the crop in a hydroponic setup;Less water consumption, leading to less power consumption;The involvement of the IoT in the setup makes the system automatic, thereby reducing human intervention.

Soil being the sustainer not only of plants but many other living things is not devoid of rodents and pests. Despite all sorts of treatment methods, plants grown in soil are never infection free. Even after treatment, the estimated percentage of infected plants in the soil medium is 20–40%, which is much more than 5–15% infected plants in the case of the hydroponic method of cultivation. The complete dependency on soil for cultivation is abolished in vertical farming methods, and the farming is conducted in vertical stacks. Thus, the quantitative production increases up to 10 times compared with the substrate cultivation method.

Dependency on weather conditions also has no role to play in the hydroponic cultivation methods, thereby enabling year-round production with much better quality in terms of leaf count, leaf length, and perimeter leaf. Keeping in mind the comparative analysis of lettuce cultivation in the simulator and the actual setup, the response to RQ. 4 can be stated as the actual comparative results completely comply with those of the simulator results. The answer to RQ. 5 can be analyzed as the NFT method of hydroponic farming is proved to be more beneficial in terms of sustainability, cost-effectiveness, energy optimization, and increased productivity of romaine lettuce.

### 4.6. Comparison of Energy Consumption: Substrate Cultivation vs. Hydroponics

Energy efficiency is another major challenge in the case of vertical farming methods. Methods to cope with this issue must be worked upon in the future. Smart hydroponic setups can be built near renewable energy generation sources.

Though hydroponics offers almost 10 times more crop produce using around 13 times less water compared to the substrate cultivation method, the electricity consumption is more than 70 times higher in hydroponic cultivation, that too using renewable energy sources. This is because in hydroponic systems electricity provides the system with all the environmental and nutritional requirements externally. The sensors, microcontroller, and actuators installed in the hydroponic setup ensure CEA. All require electricity to operate. On the other hand, in the substrate medium, electricity is negligible. It is only consumed by pumps for irrigation, lights for lighting at night, and cameras installed for vigilance (if any). Figure 18 shows that the substrate cultivation method outperforms the hydroponic cultivation method in terms of energy consumption.

Therefore, keeping in mind RQ 6, despite the crop lagging in growth and leaf characteristics as well as water consumption in substrate cultivation compared to hydroponic methods, substrate cultivation is more efficient method in terms of energy consumption.

## 5. Conclusions and Future Scope

The study of several research articles dealing with various methods of vertical farming has helped gain a deep understanding of the process. Therefore, a smart NFT hydroponic setup was implemented in this study, and the crop under study was romaine lettuce. The growth process and the yield from substrate cultivation were monitored and compared with that from the NFT process. The results show that the hydroponic yield of romaine lettuce outperforms the yield from the substrate method of cultivation qualitatively and quantitatively in a sustainable and cost and resource-effective manner. To verify the results obtained in practice, the required growth conditions were set in a simulator named ’AquaCrop’, and the substrate and hydroponic yields were analyzed and compared. The results show that the hydroponic yield was much better than the yield in the substrate medium. Similar results were obtained when the yield was analyzed in the actual substrate and IoT-based hydroponic cultivation, thereby justifying the results of the experiment.

Another conclusion drawn from this research is that the hydroponic system uses almost 70 times more energy than substrate cultivation. This is identified as the major challenge in the vertical farming method, despite outperforming substrate cultivation in all other means of comparison. In the future, this work can be extended by connecting a smartphone application with the smart farming setup. The data collected by the sensors can be sent to the Cloud for analysis, and the users can remotely monitor and control the farming process. The scope of the comparative analysis for a particular crop can also be expanded by taking all three modes of vertical farming into consideration. One crop that is compatible with aeroponics, hydroponics, and aquaponics will be grown in all three modes of vertical farming, and the qualitative and quantitative analysis of the product can be carried out to determine the best mode of cultivation for that crop. Enhancing the energy efficiency of vertical farming methods is another scope to improve.

## Figures and Tables

**Figure 1 sensors-23-01875-f001:**
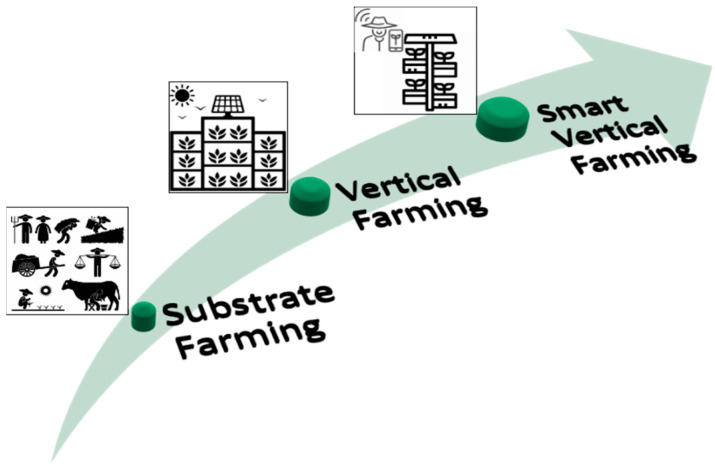
Evolution of smart vertical farming.

**Figure 2 sensors-23-01875-f002:**
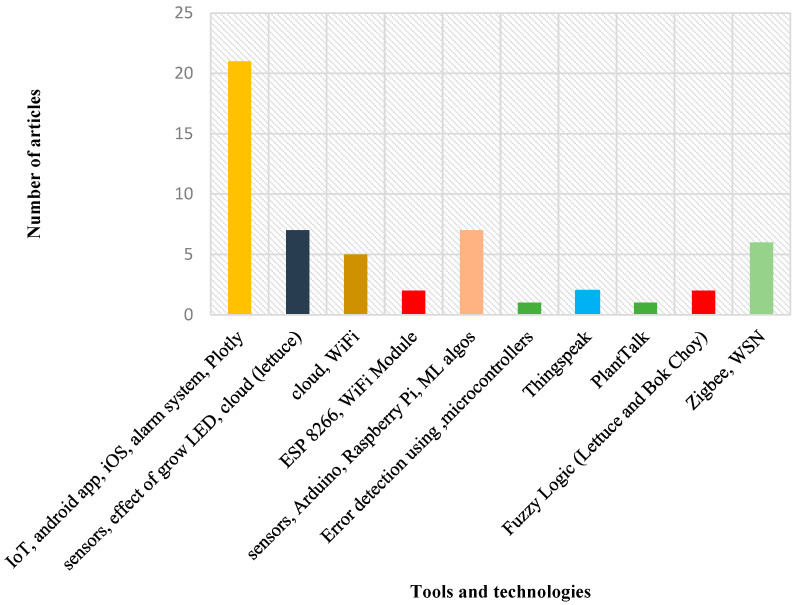
State of the art of hydroponic farming.

**Figure 3 sensors-23-01875-f003:**
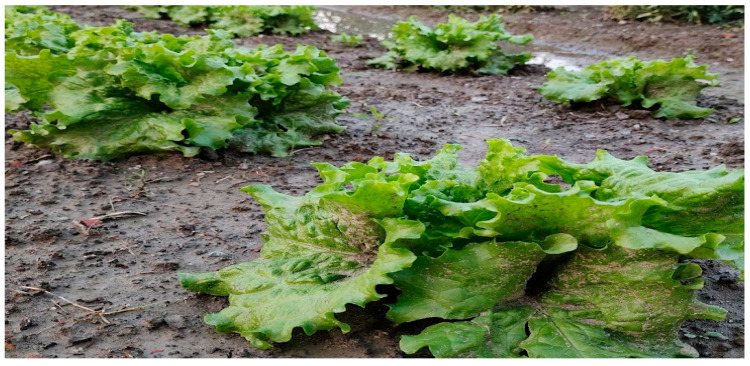
Romaine lettuce was grown in a substrate medium.

**Figure 4 sensors-23-01875-f004:**
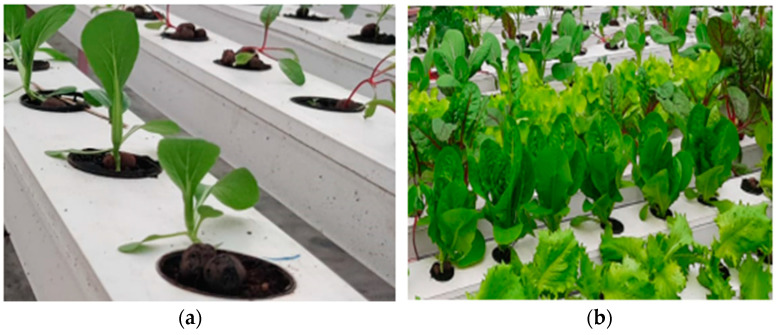
Stages of growth of romaine lettuce in the NFT: (**a**) saplings; (**b**) growing phase.

**Figure 5 sensors-23-01875-f005:**
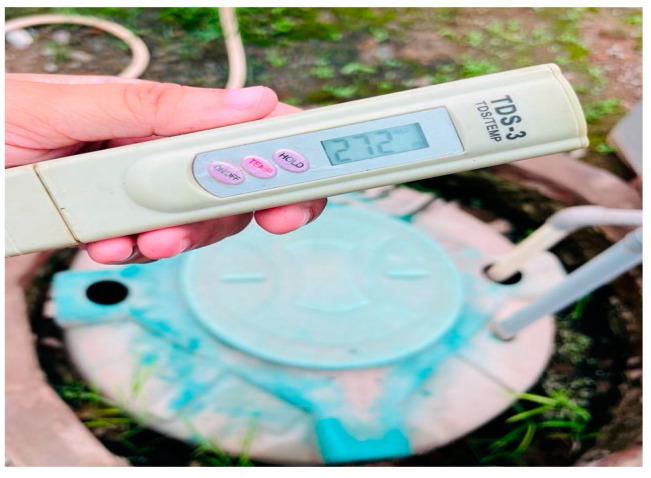
TDS measure in the solution used for the NFT.

**Figure 6 sensors-23-01875-f006:**
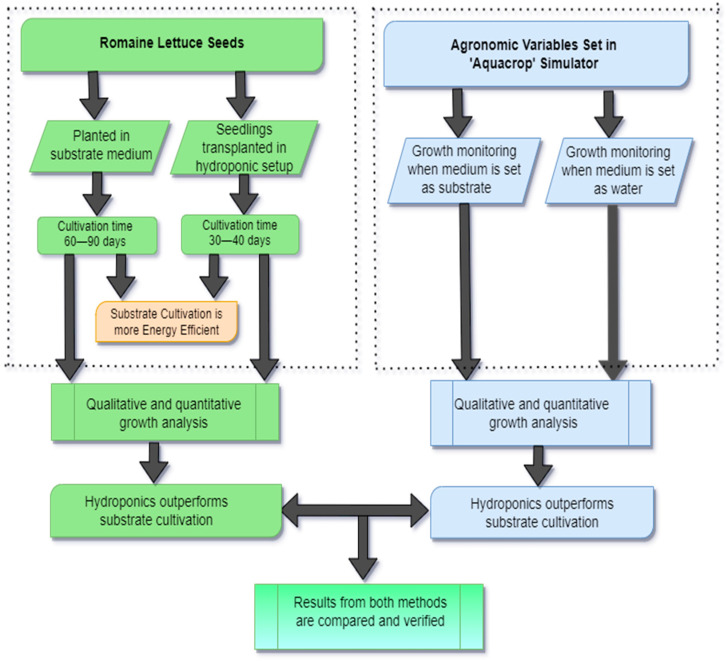
Workflow of the proposed research work.

**Figure 7 sensors-23-01875-f007:**
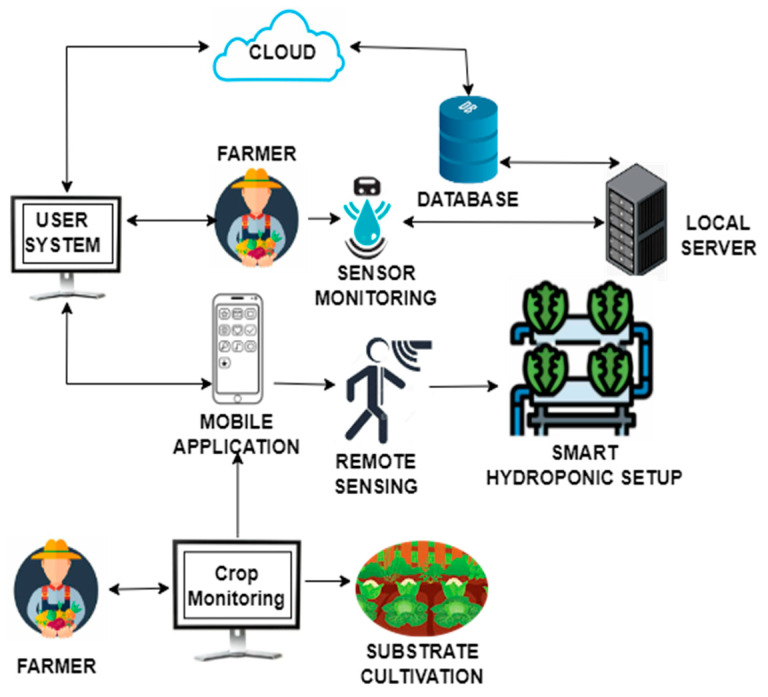
Remote sensing of smart agriculture using the IoT.

**Figure 8 sensors-23-01875-f008:**
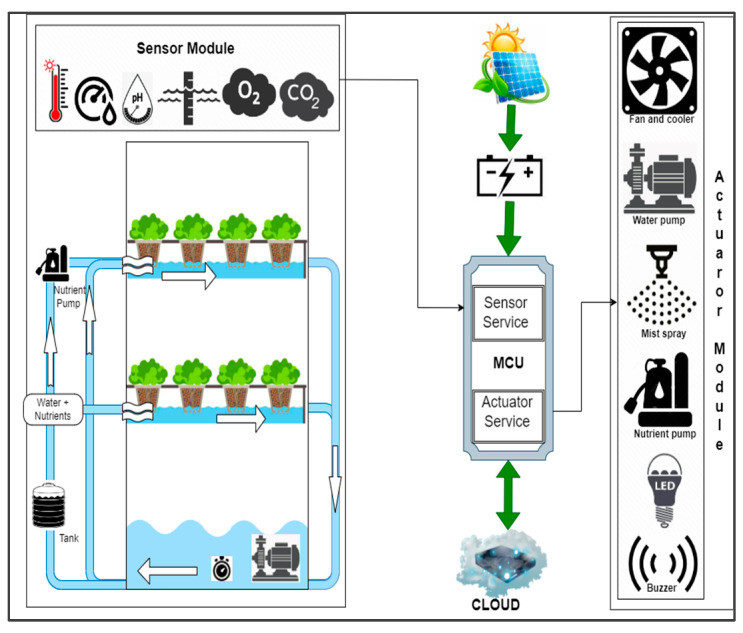
A smart hydroponic system.

**Figure 9 sensors-23-01875-f009:**
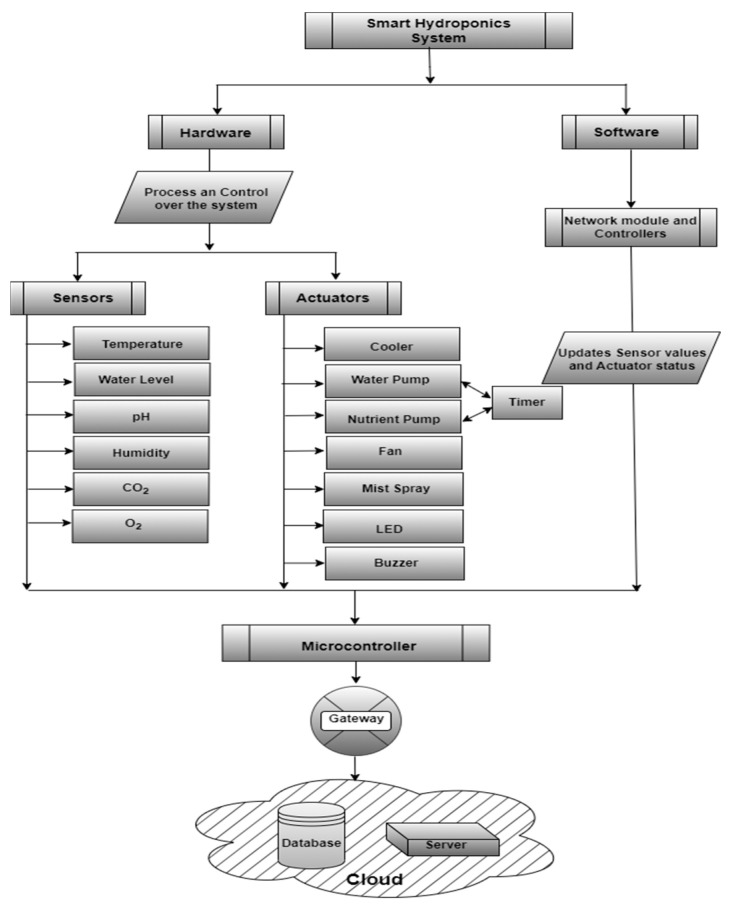
Components of the proposed smart hydroponic system.

**Figure 10 sensors-23-01875-f010:**
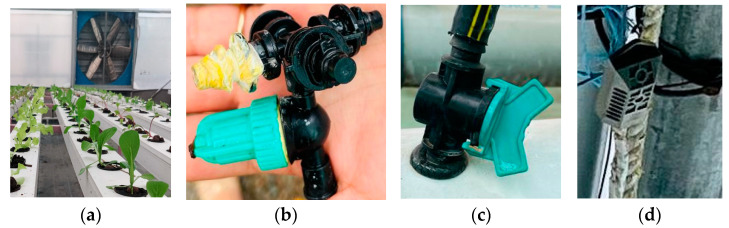
Experimental setup: (**a**) exhaust fan; (**b**) mist sprinkler; (**c**) nozzle for water flow to the grow tubes; (**d**) temperature sensor.

**Figure 11 sensors-23-01875-f011:**
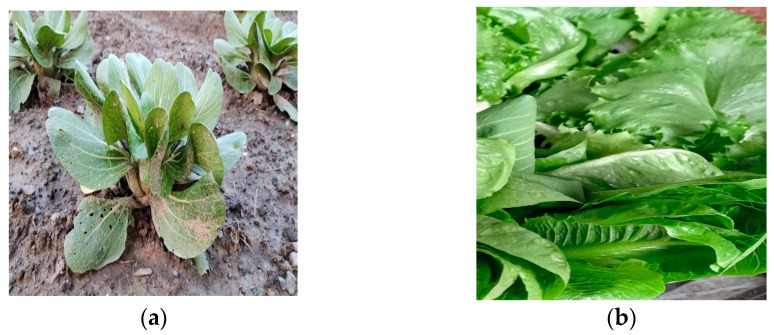
A romaine lettuce plant in different growing mediums: (**a**) an infected plant grown in soil; (**b**) a healthy plant grown in a hydroponic setup.

**Figure 12 sensors-23-01875-f012:**
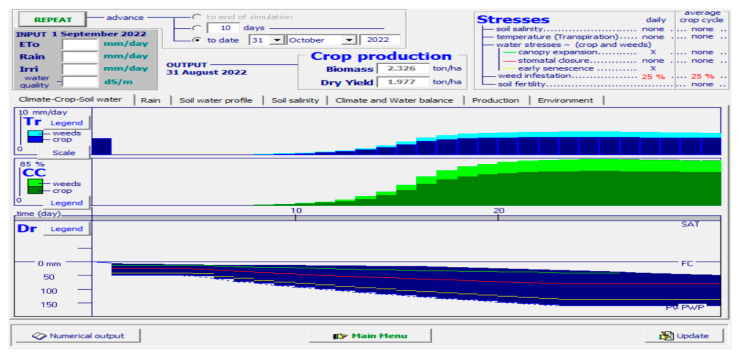
AquaCrop simulator results when fed with the conditions of romaine lettuce cultivated in soil.

**Figure 13 sensors-23-01875-f013:**
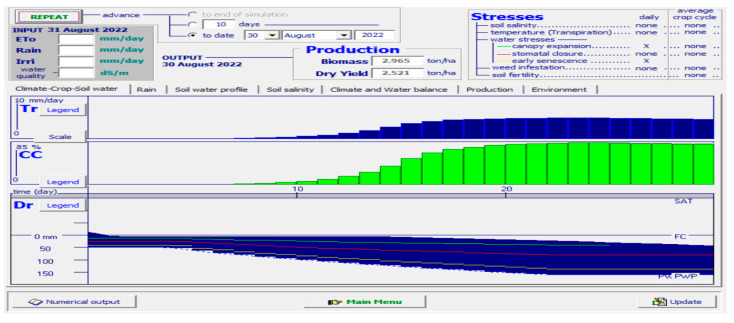
AquaCrop simulator results when fed with the conditions of romaine lettuce cultivated in hydroponics.

**Figure 14 sensors-23-01875-f014:**
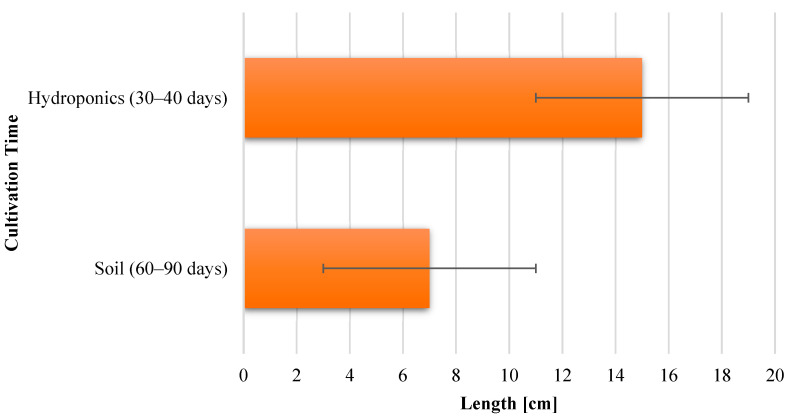
Length of fully-grown lettuce leaves in cm.

**Figure 15 sensors-23-01875-f015:**
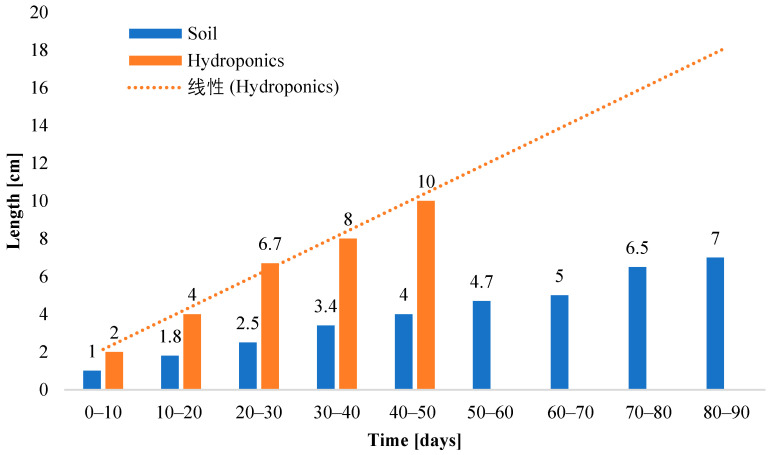
Comparison of romaine lettuce leaf length in cm grown in soil and the NFT hydroponic setup concerning time in days since sowing. A graphical representation of the periodic growth of leaf length in soil and hydroponics.

**Figure 16 sensors-23-01875-f016:**
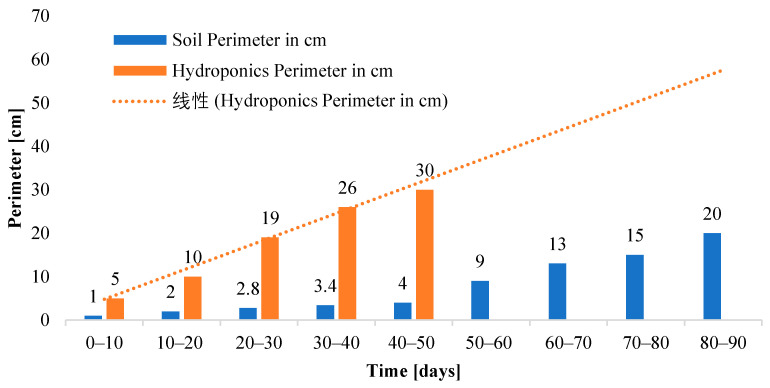
Comparison of romaine lettuce leaf perimeter grown in soil and the NFT hydroponic setup.

**Figure 17 sensors-23-01875-f017:**
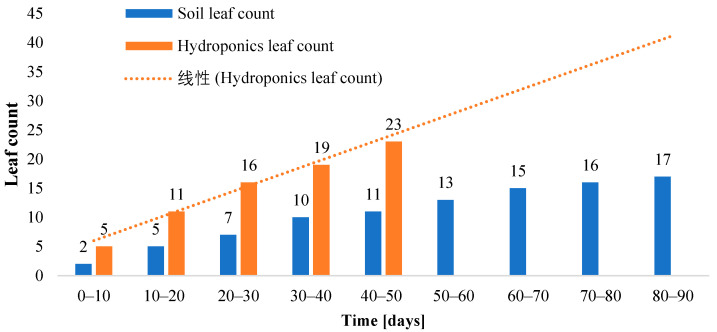
Comparison of romaine lettuce leaf count grown in soil and the NFT hydroponic setup.

**Figure 18 sensors-23-01875-f018:**
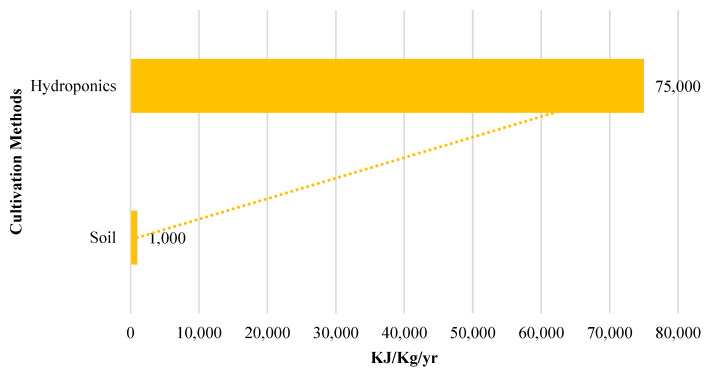
Comparison of energy consumption [KJ/Kg/yr] in the substrate and hydroponic methods of cultivation.

**Table 1 sensors-23-01875-t001:** Substrate farming vs. vertical farming.

Substrate Farming	Vertical Farming
Labor-intensive methods are involved	No labor-intensive methods are involved
The cultivation medium is soil	Soilless cultivation
Complete dependency on natural resources and climatic conditions	Completely independent of natural resources and climatic conditions
Slower crop production rate	Faster crop production rate as compared to the traditional methods
Uncertainty prevails in terms of yield and quality	Certainty in terms of yield and quality is ensured
Dependence on cattle and as a result, cattle care must be considered	No dependence on cattle at all
Large space for farming is needed	No space and no soil are needed for farming. Vertical racks are enough for cultivation with either air or water as the medium
Supports all kinds of crops	Mostly suitable for herbs and shrubs [14]
No role of technology, proper and in-depth knowledge of the crop and its nutritional, as well as agronomic requirements are required [15]	Technology plays the main role, every aspect is automated, and no knowledge of farming is needed, just the parametric ranges are to be known and set as the IoT ranges
The physical presence of the farmer is mandatory	Farmers can remotely monitor and handle vertical farms
Farming knowledge and experience are essential	Amateur farmers can work with equal efficiency

**Table 2 sensors-23-01875-t002:** Parameters and their corresponding values in the substrate cultivation method.

Parameters		Values/Ranges
Cultivation Medium		Soil—nursery bed
Cultivation Time		60–90 days
Cultivation Month		August to October
Water Consumption		260 L/Kg
pH		5.6 to 6.1
Essential Nutrient Content of Soil per Acre of Land	Nitrogen (N)	52 lb
	Phosphorous (P_2_O_5_)	350 to 420 lb
	Potassium (K_2_O)	210 lb
Other essential micronutrients include calcium (Ca), magnesium (Mg), and boron (B)		

**Table 3 sensors-23-01875-t003:** Parameters and their corresponding values in the NFT method of hydroponics.

Parameters		Values/Ranges
Cultivation Medium		Water—in a polyhouse
Cultivation Time		30–40 days
Setup		NFT Hydroponic
Cultivation Month		Possible anytime throughout the year
Water Consumption		20 L/Kg
Setup		2 Tanks–Tank A and Tank B
Tank A Content		Iron (Fe), calcium (Ca), magnesium (Mg), and 10 L stock solution
Tank B Content		MICRO–NPK stiffness phosphorous for root growth
pH		5.4 to 6.6
Nitric Acid		To reduce the pH when it exceeds 6.6
TDS		102 to 301
Temperature		25 to 30 degrees Celsius
CO_2_		<1000 ppm
O_2_		>18%
Specifications of Three Measuring Cylinders	Measuring Cylinder 1	EC 0.6–0.7, 6.2 pH
	Measuring Cylinder 2	EC 0.5, 7.4 pH
	Measuring Cylinder 3	EC 0.8, 7.4 pH
Other additional nutrient requirements include HCL and nitric acid		

**Table 4 sensors-23-01875-t004:** A comparative table gives the qualitative and quantitative analysis of the yield.

Parameters	Values/Ranges	
	Substrate Cultivation	Hydroponic Farming
Growth cycle in days	60–90	30–40
Yield in Kg/m^2^	3.9	41
Leaf Count approx.	17	23
Leaf length in cm	5–7	8–10
Leaf perimeter in cm	10–20	20–30
Percentage of infection after treatment	20%–40%	5%–15%

**Table 5 sensors-23-01875-t005:** A comparative table gives the quantitative analysis of the yield.

Parameters	Values/Ranges	
	Substrate Cultivation	Hydroponic Farming
Growth cycle in days	60	30
Biomass in ton/ha	2.326	2.965
Dry yield in ton/ha	1.977	2.521

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
