# Peer review of "Evaluation of Growth Responses of Lettuce and Energy Efficiency of the Substrate and Smart Hydroponics Cropping System"

_sensors, 2023, doi:10.3390/s23041875_

Round 1

Reviewer 1 Report

sensors- 2116102

Monitoring Growth Responses, Leaf Characterization, and Energy Efficiency in Substrate and Smart Sensors Enabled Hydroponics

Unfortunately, I have not found the scientific contribution inside this manuscript. A major rewrite is required, including some experimental results which verify simulation results.

Some points need to be known.

·     In this work, much literature is included, and the paper's objectives are unclear.

·     What are the key contributions of the proposed work? Please clear.

·    The English language needs to be improved throughout the manuscript.

·   There are no experimental results in the proposed work. Experimental results should be included for a scientific article in a serious journal.

Author Response

The authors appreciate for the comments and constructive feedback provided by the Reviewer to improve the manuscript in line with the scope and aim of the Journal.

Reviewer 2 Report

   - Kindly revise the spacing amid sentences throughout the manuscript.

   - There are also formatting problems and too many long sentences that need to be taken care of.

   - Strict guidelines of the journal should be followed precisely in the reference style.

- - There are lots of typographical mistakes in the manuscript which need to be taken care of.

Author Response

(The authors gave the same response as above.)

Reviewer 3 Report

All my suggestions and queries are in the manuscript. However, you need to consult privately an English-speaking reviewer.

Author Response

(The authors gave the same response as above.)

Round 2

Reviewer 1 Report

Sensors- 2116102

Monitoring Growth Responses, Leaf Characterization, and Energy Efficiency in Substrate and Smart Sensors Enabled Hydroponics

Thank you for allowing me to revise resubmitted manuscript titled " Monitoring Growth Responses, Leaf Characterization, and Energy Efficiency in Substrate and Smart Sensors Enabled Hydroponics " I believe the submitted manuscript and presented work are suitable for publishing in the Sensors except for one minor revision.

Minor revision: Please include the picture of the experimental setup showing smart sensors etc., used in this work. 

Author Response

The authors appreciate further for the comments and constructive feedback provided by the Reviewer to improve the manuscript in line with the scope and aim of the Journal.

Reviewer 3 Report

Please take note of lines 54-55 and 85.

Author Response

(The authors gave the same response as above.)
